# Multi-Frequency Vibration Suppression Based on an Inertial Piezoelectric Actuator Applied in Indoor Substations

**DOI:** 10.3390/mi16101178

**Published:** 2025-10-17

**Authors:** Xiaohan Li, Jian Shao, Peng Wu, Tonglei Wang, Jinggang Yang, Yipeng Wu

**Affiliations:** 1Research Institute of State Grid Jiangsu Electric Power Co., Ltd., Nanjing 211103, China; 18851790705@163.com (J.S.); 15105168844@163.com (P.W.); wangtonglei@yeah.net (T.W.); 15105168828@163.com (J.Y.); 2State Key Laboratory of Mechanics and Control of Mechanical Structures, Nanjing University of Aeronautics and Astronautics, Nanjing 210016, China

**Keywords:** inertial actuator, vibration suppression, piezoelectric actuator, substation equipment, indoor substation

## Abstract

This paper addresses the suppression of multi-frequency line spectrum vibrations during the operation of indoor substations through the development of an active vibration isolation scheme based on a piezoelectric stack inertial actuator. Based on the finite element modal analysis, the excitation frequencies that strongly influence structural response were identified, and the excitation points and sensor layout strategies were determined under a collocated control configuration. Subsequently, the actuator’s structural design and system integration were carried out. Experimental results demonstrate vibration amplitude reductions of 18.75 dB at 100 Hz, 46.02 dB at 200 Hz, and 32.04 dB at 300 Hz, respectively, validating the effectiveness of the proposed method in controlling line spectrum vibrations at multiple frequencies. The study shows that the coordinated optimization of modal matching and the dynamic response capability of inertial actuators provides experimental evidence and technical guidelines for active vibration isolation in large plate-shell structures.

## 1. Introduction

With the progress of urbanization, indoor substations are increasingly integrated into residential areas due to their low spatial cost [1,2]. Typical indoor substations are often located on the ground floor or in the basement of residential buildings [3]; the accompanying noise and vibration issues [4] therefore raise growing public concern. Such mechanical vibrations and noise are mainly caused by substation equipment [5], which also affects the structural safety of buildings [6]. Typical substation devices such as transformers [7], reactors, and fans produce periodic mechanical vibrations due to electromagnetic forces and motor imbalances [8]. Spectral characteristics of these kinds of vibrations are usually dominated by discrete line spectra centered at 100 Hz and its harmonics [9].

The reinforced concrete frame of indoor substations serves as the main transmission path for vibrational energy. Its broadband vibration response can radiate low-frequency vibrations to surrounding buildings via the support structure, potentially causing building resonance and disturbing residents. Congru et al. [10] conducted an in-depth analysis of the vibration response of the reinforced concrete frame under seismic conditions and found significant coupling effects between the equipment and the structure. Seismic isolation techniques such as lead rubber bearings (LRBs) and viscous dampers (VDs) can reduce structural accelerations significantly. However, these methods primarily target seismic frequencies below 20 Hz [11] and remain insufficient in controlling mid- to high-frequency periodic excitations. Therefore, there is a need for more frequency-specific active or semi-active control devices [12] to enhance vibration control accuracy and broaden the applicable frequency range.

Although research on noise control in urban substations is relatively comprehensive, most studies focus exclusively on noise attenuation while neglecting the transmission and control of structural vibrations. Yet, structural vibration is not only a primary source of noise but also has strong low-frequency propagation characteristics in rigid support structures, posing a potential threat to environmental quality [13]. Based on our measurement, the vibration displacement amplitude of the floor can reach as high as 5.3 μm, and the safety issue of structure cannot be ignored anymore. Fan et al. [14] implemented sound-absorbing structures, soundproof panels, and damping spring isolators in an indoor substation situated on the first floor of a residential building, achieving passive control that reduced the equivalent sound pressure level in sensitive areas on the second floor to acceptable limits. Wang et al. [15] analyzed the influence of transformer placement on vibration transmission along beams and walls in indoor substations, showing that while the vibration is strongest in a small area centered around the transformer, significant attenuation occurs between floors over long propagation distances.

Although passive vibration isolation technologies such as rubber pads and spring isolators provide limited vibration mitigation at low frequencies [16], their fixed natural frequencies and poor adaptability to multiple frequency bands limit their ability to suppress the complex vibration spectra arising from dynamic substation operations. In the vibration control of plate structures, previous studies have shown that active control methods can effectively suppress multimodal responses [17,18]. Yang et al. (2024) demonstrated that introducing dynamic inertial suspension can effectively compensate phase deviation in semi-active suspension control, highlighting the potential of inertial effects in improving system dynamic performance [19]. Active disturbance rejection control (ADRC) has been extensively investigated in piezoelectric smart structures, showing strong potential for handling uncertainties and disturbances in practical vibration control scenarios [20]. Beyond reviews, novel PIA configurations have been proposed, such as compliant-mechanism-based designs that achieve large stepping displacement and tunable load regulation, further demonstrating the structural versatility of PIAs [21]. However, most of these studies have focused on precision positioning or displacement amplification, whereas their potential for multi-frequency vibration suppression in large-scale engineering structures, such as substation equipment, remains largely unexplored. Thus, developing a multi-frequency line spectrum active vibration isolation technology for substation equipment bases is of great engineering significance. In particular, collocated control—where excitation, control, and feedback are applied at the same position—provides both practical feasibility and theoretical clarity, and serves as the main focus of this study.

## 2. Simplified Finite Element Model and Modal Analysis

To replicate the typical structural configuration of indoor substations, Figure 1 shows a scaled model. This figure presents a representative configuration, not an exact reproduction of a specific substation. Its purpose is to demonstrate features of multi-story indoor substations. In these substations, sections of the floor slabs are removed to allow installation and maintenance of large electrical equipment. This creates open spaces that span multiple stories [22]. Such “missing slab” frame configurations are commonly found in real-world applications. They often appear on the first or mezzanine floors, where slabs are removed for electrical wiring, equipment installation [23], or thermal management [24]. While this design improves spatial utilization and layout flexibility, it also introduces localized stiffness discontinuities. These changes significantly affect vibration transmission.

To reflect this feature, the scaled model, as shown in Figure 2, includes a partial floor opening on the second level, represented by a rectangular cutout to approximate this localized discontinuity, thereby illustrating the influence of the open-space design on structural vibration behavior. The distribution of electrical equipment in Figure 1 serves only as an illustrative background and was not physically included in the scaled model. Steel was selected instead of reinforced concrete to simplify fabrication and facilitate stiffness tuning. The vibration experiments aim to validate the proposed active vibration approach at several objective frequencies, which means the other performances, such as energy dissipation characteristics of the materials, are not accounted for here. However, the structural configuration and boundary conditions which influence the dynamic response are also specially designed.

This article proposes an active vibration approach based on a collocated control system, which means the excitation point, the test point and the position of the inertial piezoelectric actuator are attached to the same degree of freedom. Therefore, they are installed in a very close position, as shown in Figure 2, where points a and b are selected to observe the whole vibration level of the structure.

Moreover, the stiffness and mass distribution of the simplified model were analytically assessed and iteratively adjusted. Consequently, the modal frequencies with the experimental target frequencies (100 Hz, 200 Hz and 300 Hz) can be obtained. The goal was not to replicate real building modal frequencies, but rather to deliberately tune the model to exhibit prominent modal responses at frequencies where resonance is likely to occur, thus enabling the evaluation of vibration suppression techniques under these conditions.

As shown in Figure 3, the square opening represents the disturbed region caused by a missing floor slab. To enhance measurability and mimic actual substation base conditions, four cylindrical pillars of equal height are used as support components at the model’s base. Their geometry is based on common column foundations found in reinforced concrete substations, with thicker bases and slender upper sections to support equipment and provide isolation. During testing, the upper sections are rigidly connected to the model steel plates, while the lower sections are fixed, ensuring stability and enabling the measurement of vibrational responses at the support bases. The symmetrical geometry also helps minimize undesired eccentric responses during actuator excitation.

The bottom surfaces of the four disc-shaped legs in the boundary conditions are set as fixed constraints, while the other surfaces are free. The three-dimensional diagram of the model in COMSOL  Multiphysics 6.2 is shown in Figure 3, where the dimensions of each part are also listed, and the selected material parameters are given in Table 1. All components were meshed using free tetrahedral elements, with a total of 28,115 elements solved. The modal analyzing results are then given in Table 2. In addition, the corresponding mode shapes near the designed target frequencies are also shown in Figure 4. These simulation results indicate significant modal responses at 104.63 Hz, 199.88 Hz, and 312.30 Hz, which are close to the target frequencies of 100 Hz, 200 Hz, and 300 Hz, respectively. Given the potential errors from modeling simplifications, boundary assumptions, and numerical discretization, the results confirm the model’s frequency characteristics as reasonably accurate and suitable for engineering applications.

To facilitate theoretical investigation, simplifications were made in selecting excitation points and directions. Analyzing the modes in Figure 4, the geometric center region acts as a vibration node at 100 Hz and 300 Hz, which means that placing the actuator at the center will not effectively excite the target mode. To avoid structural holes and ensure sufficient wiring and mounting space, the excitation point was chosen slightly off-center.

In Figure 2, it can also be seen that three measurement points were established to monitor the propagation of structural response. The test point located very near the excitation position captures local response characteristics; it also provides feedback to the active control system. Test points a and b are positioned near the top and base of one supporting pillar, respectively, to observe the vibration transmission from the excitation area to the foundation.

It is worth noting that actual substation structures include irregular components such as stiffeners, brackets, and cables, which complicate idealized origin-response control. Accordingly, in the experiments described below, artificial offsets between the excitation point and actuator control point were introduced to reflect engineering practice and enhance simulation relevance.

To evaluate the response characteristics of the model within the target frequency range and verify the rationality of the excitation and measurement point settings, this paper conducted a preliminary frequency domain response analysis on the established indoor substation model. By applying a z-harmonic force (direction perpendicular to the floor) to the excitation point shown in Figure 1, the origin acceleration response (Test point), as well as the acceleration response amplitudes of test points a and b were measured, the frequency domain response curves shown in Figure 5 were therefore obtained.

It is noted that the resonance peaks obtained from the forced harmonic response analysis (104.39 Hz, 199.6 Hz, 311.6 Hz) exhibit small deviations (<0.3%) from the eigenfrequencies predicted by the modal analysis (104.63 Hz, 199.88 Hz, 312.30 Hz). Such discrepancies mainly stem from the inherent differences between eigenvalue analysis (undamped or simplified damping models) and forced harmonic response, particularly the influence of excitation and measurement positions on modal participation. These deviations fall within acceptable numerical/modeling tolerances and do not affect the subsequent control performance evaluation.

The results show that the model exhibits significant resonance response peaks near the frequency values of 100 Hz, 200 Hz, and 300 Hz, and the amplitude at the test point is the highest, while the response amplitudes at test points a and b are decrease proportionally. It indicates that the structure has considerable vibration response within the set frequency range. On the other side, this result demonstrates that the model has the basic conditions for conducting active vibration control experiments at the above frequencies. The above frequency domain response can be used as a reference benchmark for evaluating the effectiveness of subsequent control experiments, to test the suppression ability of collocated control strategies on critical frequency responses.

## 3. Design and Implementation of the Experimental Active Control Platform

### 3.1. Design of Inertial Actuator Based on Piezoelectric Stack

For the target model structure, modal analysis indicated that resonance is significant near 100 Hz, 200 Hz, and 300 Hz, and the equivalent stiffness is relatively high. Therefore, the required actuation force for vibration suppression is also relatively high, and collocated installation is adopted to maximize energy transfer. Therefore, through research and analysis, the core mortgage 100A5 piezoelectric stack actuator with displacement amplification mechanism was selected, as shown in Figure 6a. In order to increase the output displacement of the driver, a displacement amplification mechanism is added to the piezoelectric stack of the driver. The specific performance parameters of this piezoelectric stack driver are listed in Table 3. The inertial actuator structure is shown in Figure 6b; the additional mass block is connected to the piezoelectric stack actuator as a whole through long screws. Its main function is to adjust inertial mass of the designed actuator; hence the limitation of the output driving force can be changed, as well as the natural frequency of the substrate surface structure.

The piezoelectric inertial actuator is installed on the floor structure of the indoor substation by hoisting. In order to ensure sufficient output of the driving force and provide a certain pressure to the piezoelectric drive, an adjustable mass block is fixed in the direction of the driving force of the drive. When the vibration of the excitation source is transmitted to the controlled structure through the structure and actuator, the acceleration signal is picked up by an acceleration sensor (PCB352A56), which provides a feedback signal to the controller in the active control system. Then, based on the classical PID control active control algorithm, the vibration control signal is generated and served as the input of the inertial piezoelectric actuator, suppressing the structural vibration through the inertial braking force.

The dynamic model of the designed system is established under the collocated control configuration, based on the dynamic characteristics of the inertial piezoelectric actuator itself. The indoor substation model structure satisfies the following in the frequency domain.
(1)$$M_{s}s^{2}X_{s}(s)+C_{s}sX_{s}(s)+K_{s}X_{s}(s)=F_{ext}(s)-F_{a}(s).$$

Equation (1) represents the controlled structure for a single-degree-of-freedom system, where the mass, damping, and stiffness are denoted as
$M_{s},C_{s},\mathrm{and}{\;K}_{s},$ respectively.
$X_{s}(s)$ is the Laplace transform of structural displacement, and
$F_{ext}(s)$ and
$F_{a}(s)$ are the external excitation and control forces, respectively.

The sensor converts the actual acceleration into an electrical signal, which may include filtering and noise.
(2)$$Y(s)=H_{sen}(s)s^{2}X_{s}(s)+N(s),$$

The conditioned acceleration signal *Y*(s) is the result of the accelerometer’s raw electrical signal undergoing initial amplification and bandpass filtering. *N*(s) is filtered through bandpass filtering. The filtering process is employed to remove DC drift and high-frequency noise, yielding a “clean” acceleration signal relevant to the target control frequency band. Here, integration is performed to convert the measured acceleration signal into a velocity signal. The final measured velocity signal is an electrical signal that reflects the structural vibration velocity within the control frequency band, and it will be directly fed into the controller for computation. *H*_sen_(s) is the transfer function between the sensor and signal conditioning, and it represents the dynamic characteristics of the sensor. Therefore, the measured velocity signal equals the product of the true velocity and the sensor’s characteristics.
(3)$$V_{meas}(s)=\frac{1}{s}Y(s)=H_{sen}(s)sX_{s}(s).$$

The controller converts the measured signal into a control voltage.
(4)$$V(s)=K_{amp}\;G_{c}(s)\;V_{meas}(s)=K_{amp}G_{c}(s)H_{sen}(s)sX_{s}(s),$$ where the PID controller function is denoted as
$G_{c}(s)$, and *K*_amp_ is the voltage gain of power amplifier.

For actuators, if the electric displacement subsystem of the piezoelectric stack with amplification mechanism is written as
$G_{p}(s)$, then the additional mass displacement
$X_{a}(s)=\alpha G_{p}(s)V(s)$, inertial force is
(5)$$F_{a}(s)=m_{a}s^{2}X_{a}(s)=m_{a}\alpha s^{2}G_{p}(s)V(s)$$ the actuator consists of a piezoelectric stack, a displacement amplification mechanism (amplification factor
α), and an additional inertial mass
$m_{a}$.

Therefore, the closed-loop transfer function from disturbance to displacement is
(6)$$X_{s}(s)=\frac{F_{ext}\left(s\right)+m_{a}\alpha s^{2}G_{p}(s)K_{amp}G_{c}(s)N(s)}{M_{s}s^{2}+C_{s}s+K_{s}-m_{a}\alpha s^{2}G_{p}(s)K_{amp}G_{c}(s)H_{sen}(s)s^{2}}$$

This study adopts a classical PID controller to control the structural vibration. Since the positions of the excitation, the sensor and the actuator are arranged together, the PID controller is only needed to generate a braking force which is opposite to the excitation force. It is worth noting that, when the PID controller is tuned on, the designed inertial piezoelectric actuator can also output a voltage through direct piezoelectric effect. This means the phase of the control signal almost opposite to this output voltage. Therefore, the controller mainly adjusts the control signal amplitude to the inertial actuator, making sure the final driving amplitude is slightly lower than the excitation force amplitude.

### 3.2. Active Vibration Control Platform

As shown in Figure 7, the control system integrated into PSV-500 (Polytec) generates the control and drive signals. These are the input of the power amplifiers, which are connected to the inertial piezoelectric actuator and the shaker, respectively. PSV-500 is also selected to acquire the acceleration signals of the controlled structure. The major experimental platform is a steel plate connected to four cylindrical supports, as shown in Figure 7b, and the inertial piezoelectric actuator is fixedly connected to the plate through a bolt.

Because the actual indoor substation structure is made of reinforced concrete, and the practical vibration displacement amplitude of the floor is measured as 5.3 μm. The vibration acceleration at 100 Hz is therefore about 2.09 m/s^2^. Based on this measured value, we adjusted the excitation signal amplitude and made the shaker output the reasonable excitation force. This amplitude value is also selected at the other two frequencies. Finally, the active vibration control performances are measured and discussed in the following section.

## 4. Experimental Results and Discussion

Figure 8, Figure 9 and Figure 10 show the vibration control results of the steel structure using the designed inertial piezoelectric actuator, where the excitation frequencies are 100 Hz, 200 Hz, and 300 Hz, respectively.

Subfigures (a) in Figure 8, Figure 9 and Figure 10 plot the time-domain vibration acceleration waveforms of the structure before and after the control. It can be observed that the steady-state vibration amplitude is significantly reduced once the active control system is turned on.

Subfigures (b) in Figure 8, Figure 9 and Figure 10 zoom in on the corresponding waveforms from the marked time. The upper panel shows the unfiltered signal prior to control, while the lower panel shows the signal after control. Although high-frequency noise remains in the raw measurements, the amplitude of the periodic vibration is clearly attenuated after control.

Subfigures (c) in Figure 8, Figure 9 and Figure 10 show the frequency-domain responses of the vibration acceleration before and after the control. The peak frequencies are the corresponding excitation frequencies. It is also clearly seen that the peak values are significantly decreased, confirming that the proposed inertial piezoelectric actuator with PID controller can effectively suppress the structural vibrations at the target frequencies.

According to subfigures (c), we will find

(1)The acceleration amplitude decreases from 2.34 m/s^2^ to 0.27 m/s^2^ at 100 Hz;(2)The acceleration amplitude decreases from 4.00 m/s^2^ to 0.02 m/s^2^ at 200 Hz;(3)The acceleration amplitude decreases from 0.40 m/s^2^ to 0.01 m/s^2^ at 300 Hz.

Consequently, the acceleration attenuations at frequencies of 200 Hz and 300 Hz are better than that of 100 Hz. Based on the modal analysis results mentioned above, it can be concluded that these two frequencies are close to the typical modal frequency of the structure, and the structure has high dynamic response sensitivity near this frequency. Therefore, the control input of the inertial actuator can more effectively act on the structure and suppress the vibration. In contrast, the control performance at 100 Hz is slightly inferior, this is because the actuator location is close to the nodal region of the modal shape, which theoretically reduces actuation efficiency. However, collocated installation near the excitation point ensures that suppression remains achievable, as the actuator is not positioned exactly at the nodal line.

In addition, the response characteristics of the control system under different frequency bands may also affect the final control effect. The output power of piezoelectric stack normally increases with frequency, which means it has better performance at higher operation frequency, thereby achieving more significant vibration suppression at 200 Hz and 300 Hz frequencies in this experiment.

## 5. Conclusions

Based on the above experimental results, the inertial actuator can effectively weaken the resonance response of the platform structure at the typical modal frequency of the structure, proving its feasibility for active control under harmonic excitation. Especially at 200 Hz and 300 Hz frequencies, the vibration attenuation can reach as high as 40 dB. At the frequency of 100 Hz, the control effect is weaker because the actuator location is close to the nodal region of the 104.63 Hz mode, which theoretically reduces actuation efficiency. Nevertheless, the experimental result confirms that the vibration attenuation is about 19 dB, demonstrating the effectiveness and the robustness of the collocated control system.

In practical substation applications, the experiment results imply that placing inertial actuators underneath the equipment base yields the best control effectiveness. In confined spaces such as indoor substations, the active control systems can effectively intervene in the resonance generated by equipment bases or support structures without changing the original structure of the equipment, reduce noise source intensity, and improve the operation and maintenance environment.

However, there are still certain limitations to the current experiment. Firstly, the adopted structure is a scaled model that ignores the additional mass of the substation equipment on the model, and does not fully consider the load disturbance and complex boundary conditions under the operating conditions of the reactor equipment, and its equivalence needs further verification. Secondly, the control system mainly relies on single frequency excitation and lacks control evaluation under multi frequency or even wideband random disturbances, making it difficult to fully reflect its actual engineering performance. In addition, the control strategy is relatively basic and has not introduced more intelligent adaptive or model predictive control algorithms. There is still room for improvement in the control effect of some frequency bands. Further in-depth research is needed in the future.

## Figures and Tables

**Figure 1 micromachines-16-01178-f001:**
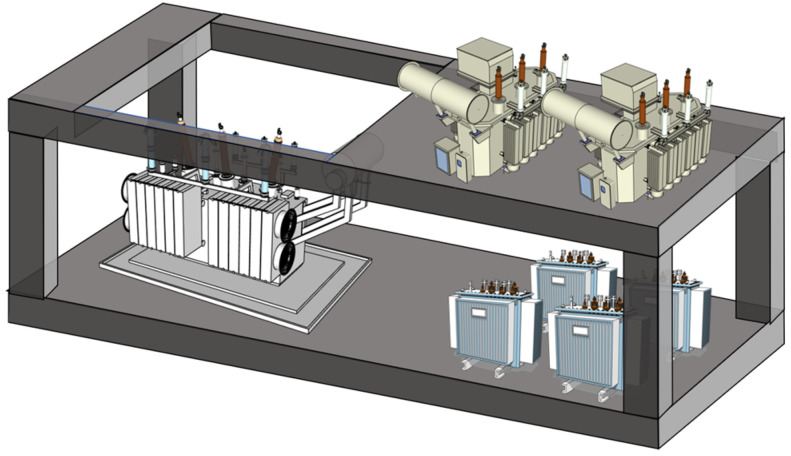
Schematic diagram of indoor substation.

**Figure 2 micromachines-16-01178-f002:**
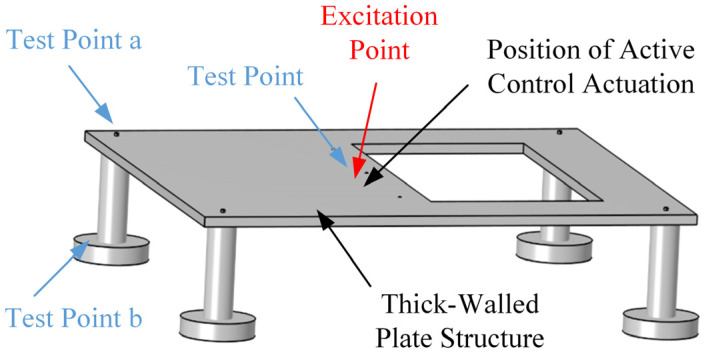
Simplified research model of indoor substation.

**Figure 3 micromachines-16-01178-f003:**
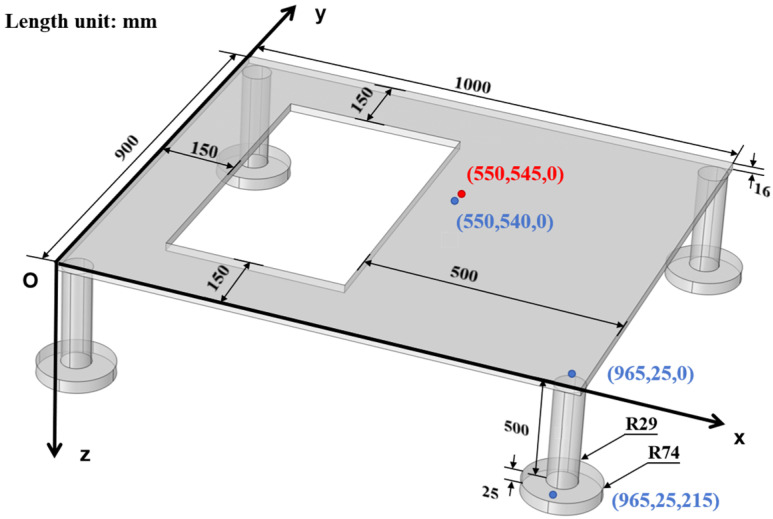
Dimensions of the equivalent floor model of the indoor substation.

**Figure 4 micromachines-16-01178-f004:**
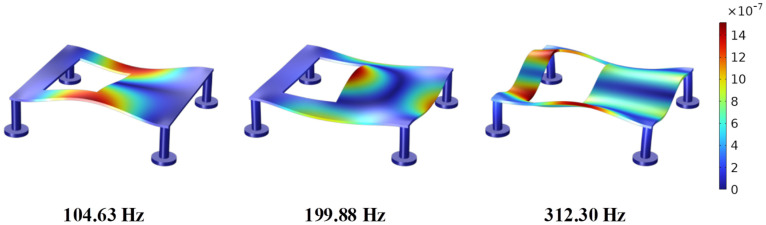
Modal shapes and frequencies of the simplified model.

**Figure 5 micromachines-16-01178-f005:**
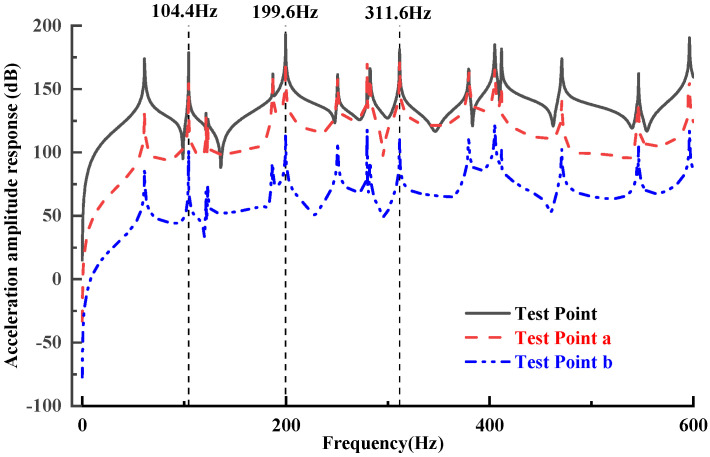
Simulated frequency response curves of the model.

**Figure 6 micromachines-16-01178-f006:**
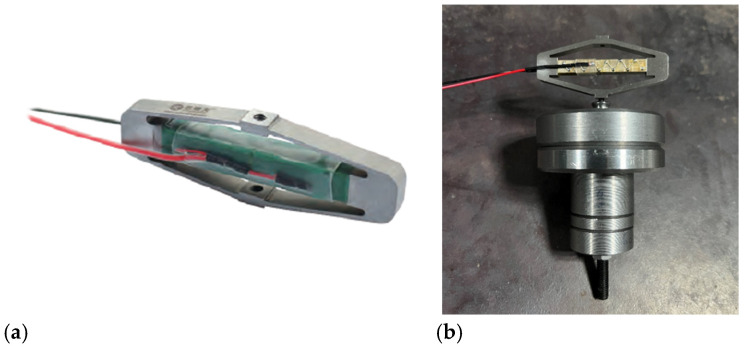
Photo of the inertial piezoelectric actuator: (**a**) core mortgage 100A5 piezoelectric stack actuator with displacement amplification mechanism, (**b**) piezoelectric inertial actuator.

**Figure 7 micromachines-16-01178-f007:**
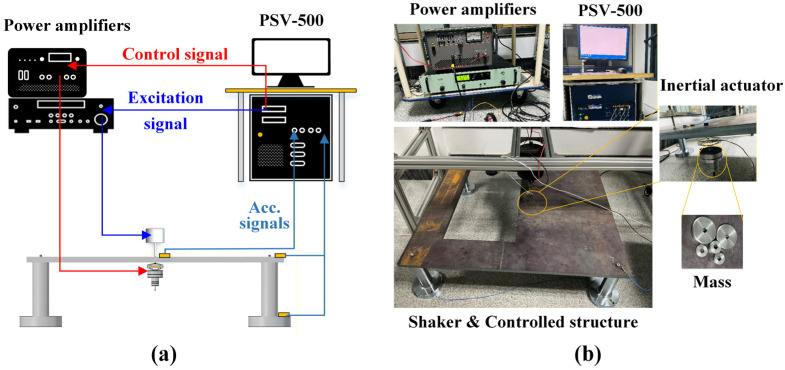
(**a**) Schematic diagram of the experimental platform, (**b**) Photos of equipment.

**Figure 8 micromachines-16-01178-f008:**
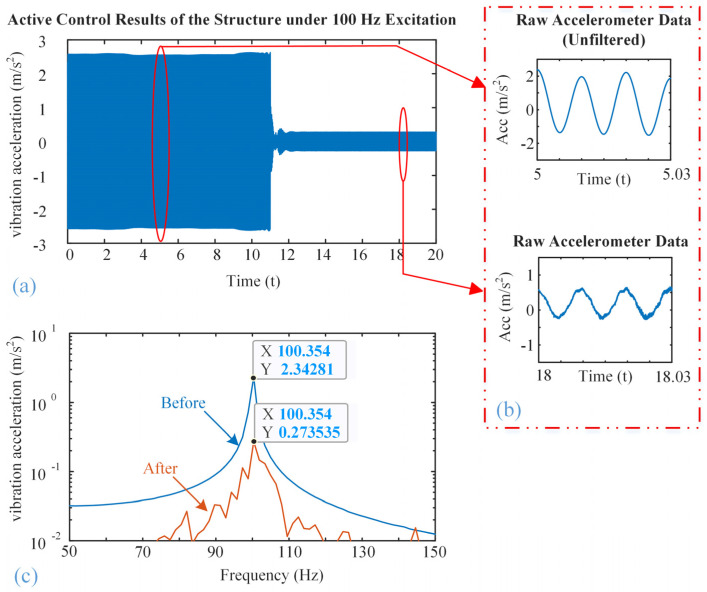
Active vibration control effect at the excitation frequency of 100 Hz: (**a**) Time-domain waveform, (**b**) zoomed waveforms, (**c**) amplitude frequency response curves.

**Figure 9 micromachines-16-01178-f009:**
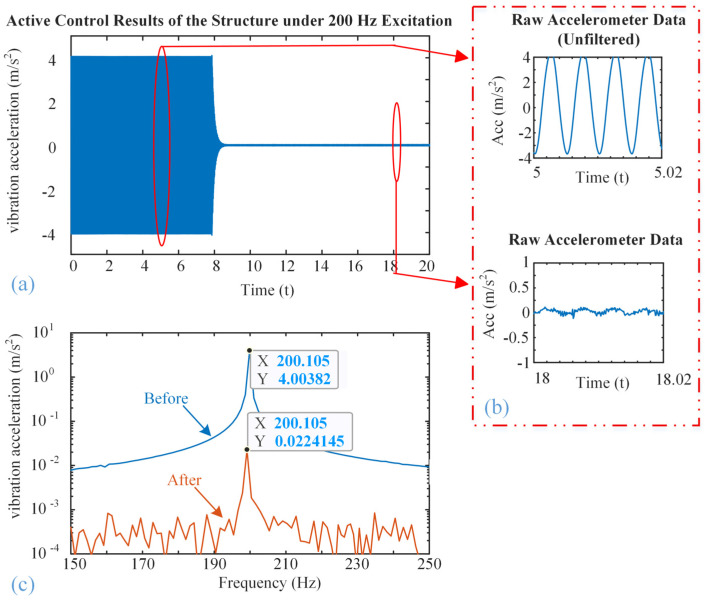
Active vibration control effect at the excitation frequency of 200 Hz: (**a**) Time-domain waveform, (**b**) zoomed waveforms, (**c**) amplitude frequency response curves.

**Figure 10 micromachines-16-01178-f010:**
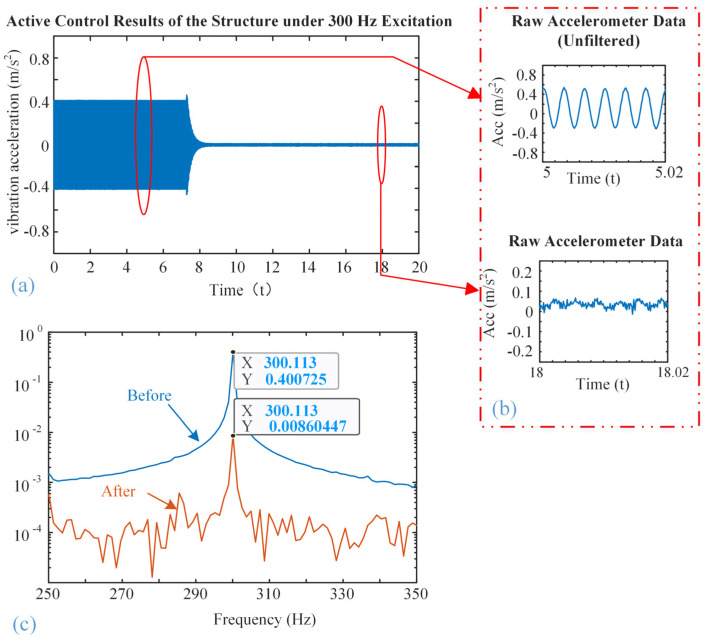
Active vibration control effect at the excitation frequency of 300 Hz: (**a**) Time-domain waveform, (**b**) zoomed waveforms, (**c**) amplitude frequency response curves.

**Table 1 micromachines-16-01178-t001:** Material parameters.

Property	Steel
Young’s modulus (Pa)	2 × 10^11^
Poisson’s ratio	0.3
Density (kg/m^3^)	7850
Material loss factor	0.001

**Table 2 micromachines-16-01178-t002:** Modal frequencies of the model.

Mode	Frequency (Hz)	Mode	Frequency (Hz)
1	61.49	7	199.88
2	104.63	8	251.21
3	121.92	9	280.10
4	123.63	10	283.30
5	186.55	11	312.30
6	187.47	12	380.35

**Table 3 micromachines-16-01178-t003:** Performance parameters of core morrow 100A5 piezoelectric stack drivers.

Parameter		Parameter	
Drive voltage	0~150 V	Equivalent electrostatic capacity	3.6 μF
Output displacement	0~100 μm	No-load resonant frequency	1900 Hz
Equivalent stiffness	2.1 N/μm	Output in the direction of motion	0~210 N

## Data Availability

Dataset available on request from the authors.

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
