# Peer review of "Multi-Frequency Vibration Suppression Based on an Inertial Piezoelectric Actuator Applied in Indoor Substations"

_micromachines, 2025, doi:10.3390/mi16101178_

Round 1

Reviewer 1 Report (Previous Reviewer 1)

Comments and Suggestions for Authors

The revised article has undergone significant improvement.

Author Response

We sincerely thank the reviewer for the positive assessment of our revised manuscript and for the recommendation for acceptance.

Reviewer 2 Report (New Reviewer)

Comments and Suggestions for Authors

The authors of this manuscript develop a multi-frequency vibration suppression based on an inertial piezoelectric actuator for indoor substations. Initially a finite element analysis modeling was carried out to determine excitation frequencies and sensors placement strategies under a collocated control configuration and performed optimization. Then experimental verification of the output characteristics of the vibration suppression based on an inertial piezoelectric actuator was conducted. It is great work in the field of piezoelectric actuators for vibration suppression and well-presented article. However, the manuscript should be revised with minor changes before it can be published. Please check the grammar and English again. It is strongly recommended to obey the rules of academic writing. My comments are followings.

  1. Could you please mention the units in Figure 3.
  2. For cylindrical pillars, the authors used concrete material for FEA simulations while in experimental prototype its seems like stainless steel, why different and provide material parameters for pillars material.
  3. In FEM what was the mesh type and size, how many elements were solved? Provide more information about boundary conditions, performed study type, and in the heading (section 2) you may add finite element modeling.
  4. Why did the authors choose test points on one supportive pillar?
  5. Is it displacement or stress or stiffness shown in figure 4, please mention the parameters. What was the input for these model excitations? Please clarify for readers.
  6. Modify Figure 7 by providing a complete graphical illustration/schematic workflow diagram of all the devices and prototype used along with their connectivity etc for better understanding of experimental setup for readers.
  7. Figure 8 experimental platform picture should also need to be modify to show all the equipment’s used in the experiment including shaker, accelerometer, power amplifier etc..
  8. Provide more detailed information on experiments including all input and output parameters values and complete experimental steps and procedures. How much accelerations was applied by shaker, power amplifier input output etc

Author Response

We sincerely thank your careful review work on our manuscript. According to your nice suggestions, we tried our best to improve the manuscript once again, the corrections were also marked in red in the re-submitted version. Appended to this letter is our point-by-point response to your comments (Please see the attachment). We hope that the new version will meet with your approval.

Reviewer 3 Report (New Reviewer)

Comments and Suggestions for Authors

In my opinion, the paper presents a simplified model of a table with an additional element featuring a rectangular cutout. While this is a practical example that may be encountered in everyday applications, the manuscript lacks a thorough analysis of the obtained results and does not attempt to optimize the design or performance. Although both numerical and experimental results are compared, the numerical model — developed in COMSOL — is not described in sufficient detail. A proper explanation of how the model was constructed, including the boundary conditions and simulation parameters, is essential. Simply stating that the simulations were performed in COMSOL and reporting the resulting frequencies is insufficient for scientific rigor.

Therefore, I can't recommend it for publication. 

My detail suggestion are:

1. Chapter 2 begins abruptly with Figure 1, which disrupts the logical flow of the manuscript. It would be more appropriate to first introduce the model conceptually before presenting any figures.

2.  The introduction section is overly general. References [3–6] are grouped into a single sentence concerning vibrations, which weakens the scientific grounding. Numerous unrelated works could be cited in this manner, but citations should be reserved for studies that are directly relevant and substantively connected to the topic.

3. I recommend revising this section to include references that are more closely aligned with the subject of the paper. Although I see that the manuscript has undergone previous review and some improvements have been made to the introduction, in my opinion, this part still lacks depth and specificity.

4. The description of the model is crucial. The potentail readres can allows repet the exreriment/simulation. Please add detailed information about coordinates test point A, B and excitation point. THis is not clear from Fig. 2

5. What is the two dots in Fig. 3? 

6. What does it mean that the small equipment mass was omitted? What qualifies as "small mass"? Is it 0.01 kg, 0.1 kg? Mass plays a crucial role in dynamic analysis, especially due to its influence on resonance behavior. On what basis did the author assume that this mass is negligible?

7. More detailed information about the COMSOL model should be provided, including which modules were used, boundary conditions, mesh type and size, etc. Please consider adding an image of the mesh model.

8. Why are there no stress concentrations in the corners of the cutouts? Logically, such regions should exhibit elevated stress levels.

9. Once again, Chapter 3 begins with a figure. Please ensure that each chapter starts with introductory text before presenting figures.

10. How does the mass of the inertial actuator affect the obtained results?

11. What methodology was used to select test points A and B? It seems that the plate would have been a better choice for excitation, allowing for more excitation points.

12. How was the position of the inertial actuator determined? Could its placement be optimized?

13. Please explain Equation (3) in more detail, particularly the band-pass filtering process.

14. What is the parameter K_amp? Please define and explain its role in the analysis.

15. In my opinion, the control algorithm is not described clearly. Please improve the explanation and provide more detail.

Comments on the Quality of English Language

The English should be reviewed and improved by a native speaker.

Author Response

We sincerely thank your professional review work on our manuscript. According to your nice suggestions and incisive comments, we tried our best to improve the manuscript, the corrections were also marked in red in the re-submitted version. Appended to this letter is our point-by-point response to your comments (Please see the attachment). We hope that the new version will meet with your approval.

Round 2

Reviewer 3 Report (New Reviewer)

Comments and Suggestions for Authors

The paper has been strong revised according to all my suggestions. Therefore, I can recommend it for publication, even though I was initially opposed to it.

This manuscript is a resubmission of an earlier submission. The following is a list of the peer review reports and author responses from that submission.

Round 1

Reviewer 1 Report

Comments and Suggestions for Authors

This paper addresses the suppression of multi-frequency line spectrum vibrations during the operation of indoor substations through the development of an active vibration isolation scheme based on a piezoelectric stack inertial actuator. Some comments should be considered:
1. Figure 1 is recommended to devide into two figures. And how can we neglect the mass of the equipments in the left figure?
2. Figure 3 should add the color bar to show more of the details.
3. Figure 4, the test point, test point a, and test point b are advised to be tagged Figure 1 or 2.(although have in Figure 6)
4. Regarding the multi-frequency vibration suppression, some references are absent(10.1007/s10409-023-23367-x.). 
5.  It is recommended to incorporate references from the past three years to enhance the innovative aspects of this paper.
6. Figure 5, does there have some dynamic model established about the piezoelectric inertial actuator?
7. Figure 8 is advised to be tagged with (a), (b), and (c) for the paper's readability.

Reviewer 2 Report

Comments and Suggestions for Authors

Review on: Multi-Frequency Vibration Suppression Based on An Inertial Piezoelectric Actuator Applied in Indoor Substations (micromachines – 3802216)

The manuscript describes an inertial piezoelectric actuator for active suppressing multi-frequency line spectrum vibrations during the operation of indoor substations. Based on experimental results obtained with a scaled model, the authors demonstrate effective vibration amplitude reductions with their proposed actuator system. The manuscript is structured into 4 main chapters. The introduction provides the reader with the necessary information on the topic, however, the state-of-the-art should be extended (see below). The design of the actuator and the control scheme are described in sufficient detail. The experimental setup and methods used are described as well. Results are presented and discussed in sufficient detail. Though, active vibration control is not a new technology and a lot of research and development has been done in the last decades. The novelty and impact of the manuscript with the described concept of the vibration reduction system is not clear. The state-of-the-art in the introduction section does not reflect the work that has been done in this area. There are a lot of review papers on the topic, which should be included in the manuscript.

Main chapter 2 is quite long. For better clarity, chapter 2 could be divided into two main chapters having section 2.1 as a separate main chapter.

For the reader it is not clear, what the advancement of the proposed vibration suppression including an actuator and a control scheme in comparison to the state of the art is. Which performance improvements have been achieved? What is the novelty in comparison to the state-of-the-art?

Reviewer 3 Report

Comments and Suggestions for Authors

See the attached paper.
